# New Perspectives on the Use of Resveratrol in the Treatment of Metabolic and Estrogen-Dependent Conditions Through Hormonal Modulation and Anti-Inflammatory Effects

**DOI:** 10.3390/cimb47090692

**Published:** 2025-08-27

**Authors:** Guilherme Renke, Ana Carolina Fuschini, Beatriz Clivati, Laura Mocellin Teixeira, Maria Luisa Cuyabano, C. Tamer Erel, Eliane Lopes Rosado

**Affiliations:** 1Nutrindo Ideais Performance and Nutrition Research Center, Rio de Janeiro 22411-040, Brazil; carolfuschini@gmail.com (A.C.F.); beatriz.clivati@hotmail.com (B.C.); lauramocellinteixeira@gmail.com (L.M.T.); m_luisavieira98@hotmail.com (M.L.C.); 2Programa de Pós-Graduação em Nutrição, Instituto de Nutrição Josué de Castro, Universidade Federal do Rio de Janeiro, Rio de Janeiro 21941-902, Brazil; elianerosado@nutricao.ufrj.br; 3Department of Obstetrics and Gynecology, Cerrahpasa School of Medicine, Istanbul University, Istanbul 34320, Turkey

**Keywords:** trans-resveratrol, estrogen receptor, metabolic syndrome, anti-inflammatory

## Abstract

Estrogen-dependent conditions, such as endometriosis, adenomyosis, lipedema, polycystic ovary syndrome, and breast cancer, are intimately involved with hormonal changes related to estrogen and their receptors. These conditions can be expressed mainly during hormonal changes such as pregnancy, puberty, and menopause. They are associated with alterations in estrogen function and inflammatory mechanisms, leading to significant discomfort and a marked decrease in self-esteem in women. Resveratrol has been studied in the treatment of inflammatory diseases like obesity, metabolic syndrome, and endometriosis. The research suggests potential pathways through which resveratrol may also be beneficial in treating metabolic and estrogen-dependent conditions. We reviewed 63 articles from 2000 to 2025, prioritizing systematic reviews, meta-analyses, and randomized controlled trials in the PubMed, ScienceDirect, and SciELO databases. Our results suggest that resveratrol may benefit metabolic and estrogen-dependent conditions by modulating anti-inflammatory factors that regulate estrogen receptor activity, increasing lipolysis, decreasing insulin resistance, and mitigating oxidative stress. Future research should evaluate the long-term safety and potential therapeutic effects of resveratrol in metabolic conditions.

## 1. Introduction

Estrogen is a key hormone involved in numerous physiological and pathological metabolic processes. Beyond its fundamental role in the reproductive system, estrogen also plays a central part in the pathogenesis of several diseases, including endometriosis, adenomyosis, lipedema, polycystic ovary syndrome (PCOS), and breast cancer (Figure 1). These conditions are not typically characterized by altered circulating estrogen levels but rather by estrogen receptor (ER) hyperactivity, which is frequently associated with a significant genetic predisposition [1,2]. Given their higher prevalence in women, estrogen-dependent disorders usually manifest during periods of hormonal fluctuation such as puberty, pregnancy, and menopause [1,2,3]. Their nonspecific clinical presentation often contributes to underdiagnosis and treatment delays, thereby increasing morbidity and exerting a negative impact on quality of life, mental health, and overall functioning in affected women [3].

Moreover, these conditions frequently coexist or share pathophysiological mechanisms with metabolic diseases. In 2021, obesity represented the leading cause of disability-adjusted life years (DALYs) among women of reproductive age, followed by hypertension and type 2 diabetes (T2DM) [4]. It is estimated that metabolic syndrome currently affects approximately 30% of the global population, with projections suggesting an increase to over 50% in the coming decades [5]. Postmenopausal estrogen deficiency further exacerbates metabolic disturbances, promoting visceral adiposity, insulin resistance, T2DM, and cardiovascular diseases [6]. Estrogen has been recognized for its regulatory role in energy metabolism, glycemic homeostasis, and chronic metabolic inflammation, making it a potential therapeutic target in metabolic and inflammatory disorders [7,8]. In this context, the development of early diagnostic strategies, individualized therapeutic approaches, and public health policies is crucial for the prevention and management of these interrelated conditions, which disproportionately affect the global female population.

Resveratrol has garnered increasing attention as a promising therapeutic compound for these diseases due to its diverse bioactive properties [9], including antioxidant, anti-inflammatory, and estrogen-modulating effects [7]. Resveratrol is a phenolic stilbene primarily found in plant-based foods, existing in various concentrations and chemical forms, namely cis and trans isomers, which may occur either glycosylated (as glucosides) or in free form (as aglycones), the latter being more bioavailable (Figure 2) [10].

According to data from the Spanish EPIC cohort, more than 98% of dietary resveratrol intake is derived from red wine, which may contain up to 1.979 µg/mL of trans and cis aglycones [11]. High concentrations of trans-resveratrol glucoside are also found in the root and leaves of the itadori plant (polygonum cuspidatum), with levels ranging from 0.342 µg/mL to 974 µg/mL. Other relevant dietary sources include grapes (in the trans-glucoside form), peanuts, and berries (in the aglycone form) (Table 1). The content of resveratrol varies depending on plant species, climate, geographic location, and processing methods [7]. Regarding its isomeric forms, cis- and trans-resveratrol exhibit distinct biological activities. The trans form can isomerize to the cis configuration upon exposure to ultraviolet light, influenced by factors such as temperature, pH, and duration of irradiation, thereby reducing its biological activity due to its molecular configuration, in which functional groups are aligned on opposite sides of the molecule [12]. The cis form is less studied, as it is rarer in nature and considered to have lower antioxidant potential compared to trans-resveratrol, which has been more extensively investigated for its antioxidant, antiproliferative, and pro-apoptotic properties [9,12].

The trans isomer offers several advantages, including its abundance in natural sources such as red fruits, where molecular groups are positioned on opposite sides, thereby enhancing its bioavailability. Trans-resveratrol inhibits tubulin-associated unit (TAU) protein aggregation, which is related to tubulin and interferes with the production, aggregation, and clearance of Aβ protein, a process involved in neuroinflammation. It also regulates cellular metabolism, cholesterol metabolism, and the expression of apolipoprotein E4 (APOE4) [13].

Resveratrol exhibits both antioxidant and metabolic activity, largely attributed to the presence of hydroxyl groups. Their distribution in the aromatic rings of its stilbene structure confers structural similarity to endogenous and synthetic estrogens, including 17β-estradiol (E2) and diethylstilbestrol. As such, resveratrol interacts with ERs, functioning as a mixed agonist/antagonist. Its phytoestrogenic activity has been demonstrated in several in vitro and in vivo models, including ER-positive breast and endometrial cancer cell lines [14].

Additionally, resveratrol exhibits anti-inflammatory effects and may benefit patients with estrogen-dependent disorders by regulating angiogenesis in associated diseases, enhancing antioxidant defense, and modulating lipid metabolism, thereby preventing the formation of atherosclerotic plaques. It also acts on endothelial function, cardiac cell apoptosis, and the expression of inflammatory mediators [15,16,17,18,19].

Despite the potential application of resveratrol as an adjuvant in the treatment of chronic and inflammatory diseases—such as T2DM, irritable bowel syndrome, Alzheimer’s disease, and cardiovascular disorders—robust evidence on the oral bioavailability of trans-resveratrol remains lacking [2]. Preclinical studies comparing oral encapsulated formulations have demonstrated that a natural self-emulsifying hydrogel of trans-resveratrol produced via uniform impregnation of resveratrol micelles into a fenugreek galactomannan hydrogel matrix, forming a water-soluble micelle/hydrogel powder, significantly prolongs its half-life (approximately 37.9% retention at pH 6.5 and 28.5% at pH 2.0 after 24 h). In contrast, conventional resveratrol exhibits minimal stability under gastric and intestinal pH, making this hybrid formulation a promising alternative [20].

Due to resveratrol’s inherent limitations—poor aqueous solubility, rapid hepatic metabolism, and low oral bioavailability—nanotechnological strategies have been proposed to enhance its therapeutic potential. Nanoparticles, typically under 500 nm in size, function as colloidal drug carriers that improve solubility, stability, tissue permeability, and pharmacokinetics. Various nanocarrier systems have been investigated for resveratrol delivery, including liposomes, polymeric nanoparticles (e.g., PLGA), solid lipid nanoparticles (SLNs), polymeric micelles, dendrimers, mesoporous silica nanoparticles (MSNs), and carbon nanotubes—each with distinct physicochemical properties [12]. These systems enable multiple administration routes—oral, transdermal, mucosal, and parenteral—broadening the clinical applications. The incorporation of resveratrol into such nanostructured carriers has demonstrated substantial improvements in molecular stability, controlled release, bioavailability, and antitumor efficacy in both in vitro and in vivo models [12]. Notably, a study using chitosan-stabilized resveratrol/selenium nanoparticles (CS/Res/SeNPs) reported superior efficacy in a murine model of T2DM—improving glycemic control, insulin sensitivity, lipid profiles, oxidative stress, inflammation, apoptosis, and organ function. The CS/Res/SeNPs-10 formulation exhibited favorable molecular interactions with components of the PI3K/AKT/mTOR pathway, reinforcing the mechanistic potential of nanoencapsulated resveratrol [21].

A growing body of evidence supports the notion that resveratrol exerts a biphasic, dose-dependent response, a phenomenon consistent with hormesis. Low to moderate doses (≤500 mg/day) tend to produce beneficial biological effects, including antioxidant, anti-inflammatory, and metabolic improvements. However, as the dosage exceeds this threshold—particularly at or above 1000 mg/day—multiple studies have reported a higher incidence of mild to moderate adverse events, predominantly of gastrointestinal origin, such as diarrhea, nausea, and abdominal discomfort. This U-shaped dose–response curve has been consistently observed across various biological systems and endpoints, including endothelial, immune, reproductive, cardiac, pulmonary, and bone cells [22,23,24,25,26].

In addition to the dose–response relationship, treatment duration appears to influence clinical outcomes. A meta-analysis of randomized controlled trials revealed that treatment periods longer than 12 weeks were associated with greater improvements in lipid profiles, suggesting a time-dependent therapeutic effect [23]. Conversely, long-term administration at high doses has been occasionally associated with mild elevations in hepatic enzymes, indicating potential hepatotoxicity, particularly after 90 days of continuous use. There are also rare reports of leukopenia, although without confirmed causality [24,25,26].

In specific populations, such as individuals with T2DM, doses between 500 and 1000 mg/day, especially when combined with oral antidiabetic agents, have been associated with mild hypoglycemic episodes, suggesting a possible synergistic effect on glucose metabolism that warrants clinical monitoring. Importantly, no serious adverse events have been conclusively attributed to resveratrol, supporting its favorable safety profile in elderly populations when used within an appropriate therapeutic window, considering both dose and duration of treatment [24,25,26]. These findings reinforce the necessity of individualized therapeutic strategies and underscore the non-monotonic nature of resveratrol’s effects.

In recent years, resveratrol has been increasingly investigated as a therapeutic agent in metabolic and estrogen-related disorders. Emerging evidence highlights its structural similarity to E2, enabling interaction with ERs and modulation of estrogen-responsive signaling pathways [27]. Metabolomic analyses by Falcone et al. (2025) demonstrated both estrogenic and anti-estrogenic activity, suggesting a dual functional role in inhibiting estrogen-sensitive cell proliferation [28]. This is especially relevant in polycystic ovary syndrome (PCOS), which involves insulin resistance and hyperandrogenism [28]. Chang et al. (2025) demonstrated that resveratrol enhances endocrine and metabolic parameters in PCOS, although the underlying molecular mechanisms require further elucidation [29]. Viana et al. (2024) found that oral resveratrol supplementation acts as a phytoestrogen, increasing estrogen levels and improving metabolic and anthropometric parameters [30]. Similarly, Brown et al. (2024) highlighted the consistent anti-inflammatory properties and beneficial effects of resveratrol on metabolic dysregulation in a systematic review [24]. Still, the authors emphasized the need for rigorously controlled clinical trials to assess its long-term efficacy and safety.

Although resveratrol shows promise for estrogen-dependent conditions, clinical data remain limited, and its use in metabolic diseases is still underexplored. It is still necessary to conduct rigorous controlled clinical trials to assess its long-term efficacy and safety. This review aims to outline the novel clinical potential of resveratrol based on its modulation of ERs and gut microbiota, as well as its anti-inflammatory, antihyperglycemic, and lipid-lowering properties.

## 2. Materials and Methods

The presented scientific literature was reviewed, and the studies were retrieved from the PubMed (https://pubmed.ncbi.nlm.nih.gov/), ScienceDirect (https://sciencedirect.com/), and SciELO (https://scielo.com.br/) databases, accessed on 30 December 2024. Combinations of several search terms—such as “trans-resveratrol”, “anti-inflammatory”, “metabolic syndrome”, and “estrogen receptor”—were applied. After the search, the studies from 2000 to 2025 were classified according to the health-specific parameters of the text and selected prioritizing systematic reviews, meta-analyses, and randomized controlled trials, resulting in a total of 63 reviewed articles. Our selected primary outcome was the antioxidant action of resveratrol. Secondary outcomes included the types, effects, and safety of resveratrol in metabolic conditions.

## 3. Results and Discussion

Estrogen-dependent conditions are increasingly prevalent, so early diagnosis and effective clinical treatment are essential to offer affected women a better quality of life and self-esteem. In the search for therapeutic options, resveratrol appears to be beneficial for this condition due to its antioxidant, anti-inflammatory, antiproliferative, and hormonal regulatory activities [14,31,32,33,34].

Resveratrol may act primarily through two pathways: activating anti-inflammatory pathways, such as sirtuins (SIRT-1) [31,32,33,34], and regulating hormones, mainly through its interaction with ERs [32]. Despite the lack of randomized controlled trials (RCTs) directly relating estrogen-dependent conditions to resveratrol, we highlight the potential of this substance to offer benefits in treating lipedema risk factors, including inflammation, insulin resistance, lipogenesis, and hormonal imbalance. Thus, resveratrol could be considered an adjuvant therapy for the control of these metabolic conditions.

### 3.1. Metabolic and Anti-Inflammatory Actions of Resveratrol

The potential metabolic effects of resveratrol can be attributed to its ability to activate sirtuins, a family of enzymes that regulate various metabolic processes related to aging, stress response, and inflammation. They are present in multiple tissues, including the brain, liver, muscle, pancreas, testes, ovaries, and adipose tissue [32].

When activated, these enzymes promote (1) the activation of factors related to mitochondrial biogenesis, such as peroxisome proliferator-activated receptor gamma coactivator 1-alpha (PGC-1α), 5′ adenosine monophosphate-activated protein kinase (AMPK), estrogen-related receptor alpha (ERRα), telomerase reverse transcriptase (TERT), mitochondrial transcription factor A (TFAM), nuclear respiratory factor 1 and 2 (NRF-1/NRF-2) [34], and peroxisome proliferator-activated receptor gamma (PPARγ) [7], which also acts as a regulator of fatty acid storage and glucose metabolism [32]; (2) modulation of phosphodiesterase enzyme activity allowing the inhibition of pro-inflammatory mediators such as interleukin-1 (IL-1), IL-6, C-reactive protein (CRP), nuclear factor kappa B (NF-kB), cyclooxygenase 1 (COX-1), and cyclooxygenase 2 (COX-2) [31,32,35]; (3) promotion of antioxidant enzyme stimulation such as superoxide dismutase, catalase, and glutathione peroxidase and reduction in reactive oxygen species formation [18]; and (4) modulation of endothelial nitric oxide synthase (eNOS) activity increasing nitric oxide (NO) production and consequent vasodilation and blood flow regulation, in addition to reducing nicotinamide adenine dinucleotide phosphate (NADPH) oxidase activity, reducing superoxide production, and preserving nitric oxide availability and endothelial function [18].

All of these actions can lead to numerous metabolic outcomes. There is evidence of some points that would be interesting for the treatment of lipedema, such as (1) reduction in lipogenesis and increase in lipolysis [32], increase in adiponectin levels, and reduction in leptin and insulin resistance [36], promotion of improvement in beta-oxidation in skeletal muscle with reduction in intramuscular fat accumulation, aid in appetite control and subcutaneous adipose tissue deposition [37]; (2) improvement of tissue fibrosis through a reduction in factors such as hypoxia inducible factor (HIF) alpha expression and a reduction in collagen deposition in adipose tissue [38]; and (3) increase in the “browning” effect in white adipose tissue (WAT) by activation of uncoupling proteins such as UCP-1, promoting thermogenesis and contributing to a reduction in adipose tissue fibrosis, as well as the improvement of its metabolism [39].

Resveratrol offers additional benefits for intestinal health by contributing to improving intestinal barrier integrity, reducing the passage of inflammatory compounds into the bloodstream, and improving the composition and diversity of the gut microbiota with increased growth of beneficial bacteria such as Lactobacillus and Bifidobacterium and a reduction in pathogenic bacteria such as Clostridia, Lachnospiraceae, and *Enterococcus faecalis*, contributing to the improvement of chronic systemic inflammation [35]. Another key point is that anti-obesity effects are promoted by resveratrol through the modulation of the “gut–adipose tissue axis” [34].

Resveratrol’s activation of SIRT may influence the expression of genes related to hormonal signaling, helping regulate the body’s hormonal response to energy metabolism [34]. There have been a few randomized clinical studies in humans, with different doses and intervention times. The outcomes studied included glycemic control, inflammatory markers, and anthropometric parameters in mice and individuals with obesity, diabetes mellitus, metabolic syndrome, and cardiovascular disease. No studies included patients with estrogen-dependent conditions, except polycystic ovarian syndrome (PCOS) (Table 2).

Resveratrol appears to have some metabolic benefits, including a reduction in intestinal oxidative stress, an improvement in gut microbiota composition and diversity, and a decrease in obesity-associated bacteria such as Desulfovibrio and Lachnospiraceae, along with an increase in the abundance of Blautia [52]. The biotransformation of resveratrol by the gut microbiota into compounds such as 4-hydroxyphenylacetic acid (4-HPA) and 3-hydroxyphenylpropionate (3-HPP) contributes to improved lipid metabolism and reduced fat accumulation in vitro [52].

Improvements in cardiometabolic risk factors such as waist circumference, average blood pressure, fasting plasma glucose, triglycerides, and HDL cholesterol, as well as biomarkers of inflammation and oxidative stress such as ultra-sensitive CRP, IL-6, TNF alpha, and malondialdehyde, were observed without significant side effects with the use of 150 mg of resveratrol/day associated with 250 mg of oral tocotrienol twice daily for 24 weeks [53]. A reduction in the expression of sterol regulatory element binding protein 1-c (SREBP1-c) and increase in the expression of peroxisome proliferated activated receptor alpha (PPARα) in the liver resulted in a reduction in lipid deposition in the liver and positively contributed to improved liver function, dyslipidemia, insulin resistance, oxidative stress, and metabolic inflammation [54].

In the case of estrogen-dependent conditions, the main hypothesized action of resveratrol can be summarized in metabolic repercussions. This involves reducing lipogenesis, increasing lipolysis, reducing insulin and leptin resistance, improving adipose tissue fibrosis, promoting the “browning” effect in adipose tissue, improving the gut microbiota pattern and intestinal barrier integrity, contributing to reducing chronic systemic inflammation, and improving the connection of the “gut–brain” axis, which aids in weight loss, ultimately promoting hormone release and regulation by reducing systemic inflammation [13,15,52,53,54,55].

Regarding estrogen receptors, resveratrol interacts with both ERs, promoting a mixed effect. It exerts a dual modulatory effect on ERα and ERβ, acting as a mixed agonist/antagonist through complex receptor interactions and structural mimicry of endogenous estrogens. Its estrogenic activity is primarily mediated by the spatial distribution of hydroxyl groups on the stilbene backbone, which facilitates its binding to the ligand-binding domains (LBDs) of both receptor isoforms in a manner similar to E2. In silico molecular docking studies have shown that resveratrol forms critical hydrogen bonds within the ERα binding site, thereby stabilizing the receptor–ligand complex and mimicking the binding conformation of E2. Experimental assays using recombinant receptors and estrogen-dependent cell lines have demonstrated that resveratrol acts as an agonist of both ERα and ERβ, although with lower potency compared to E2. At low nanomolar doses, resveratrol may stimulate proliferation in ER-positive cells, whereas at higher micromolar concentrations, it exhibits cytostatic or cytotoxic effects, likely through ER-independent mechanisms, such as the induction of oxidative stress or apoptosis. Its activity is influenced, for example, by the ERα/ERβ expression ratio, the availability of coactivators and corepressors, and its concentration-dependent hormetic behavior. This dual behavior, referred to as the “Estrogenic Paradox” of resveratrol, reinforces its classification as a selective ER modulator [14].

Given that resveratrol has beneficial effects in treating PCOS [16,17] and endometriosis [56] due to its anti-inflammatory properties, which contribute to the regulation of the hypothalamic–pituitary–ovarian axis [55], it can also promote positive hormonal regulation for new applications such as lipedema treatment through its reported antiestrogenic action in studies of endometriosis, a condition with a hormonal background similar to that of lipedema [56,57,58,59]. Thus, considering both metabolic and estrogen-dependent conditions as diseases involving alterations in body inflammatory mechanisms, resveratrol would benefit in treating these conditions.

Estrogen-dependent conditions such as endometriosis, involve increased signaling of inflammatory pathways and the influence of estrogen in their development. Studies discuss epigenetic defects that lead to increased estrogen synthesis by the aromatase enzyme, as well as abnormal and increased estrogen action via ERβ receptor in endometriosis [57,58] and, in the case of lipedema, altered distribution of ERs in adipocytes (ERα/ERß ratio) with impacts on hormone signaling, as well as increased release of steroidogenic enzymes produced by adipocytes, leading to an increase in paracrine estrogen release [59].

In this context, resveratrol has been shown to exhibit antiestrogenic activity when administered in high doses, as observed in a study where subcutaneous pellets containing 6, 30, or 60 mg of resveratrol and E2 were implanted in ovariectomized rats that had been injected with human endometrial cells [60]. This could also be beneficial for treating all other estrogen-dependent conditions. Although studies regarding the potential of resveratrol for treating these different conditions are still lacking, this substance has numerous metabolic activities that can contribute not only to improving the symptomatic aspect of the disease but also to preventing its progression to more severe, disabling conditions requiring surgical intervention.

### 3.2. Future Perspectives of Resveratrol in Estrogen-Dependent Conditions

The present study suggests that resveratrol may be an effective therapy for estrogen-dependent conditions, ameliorating these chronic, progressive, and inflammatory conditions (Figure 3). The proposed treatments for these conditions include not only hormonal therapies and surgical interventions but also several other strategies, such as weight loss, anti-inflammatory diets, and regular physical exercise. Despite the absence of studies directly linking the therapeutic potential of resveratrol to other gynecological diseases, this article reviews the literature, offering new insights into the use of resveratrol in women’s health.

Different forms of resveratrol, including the *trans* isomer (*trans*-resveratrol), are linked to improved bioavailability and antioxidant potential compared to the *cis* isomer. Future studies should focus on developing new pharmaceutical forms, particularly resveratrol formulations that enhance bioavailability and serum half-life, as these could provide significant value for future research. New molecular presentations can aid in understanding estrogen-dependent diseases.

This review suggests that resveratrol has beneficial antioxidant, anti-inflammatory, and hormonal regulatory effects. Although further studies are needed to determine the effectiveness of resveratrol in treating estrogen-dependent conditions, this study demonstrates that the substance exhibits promising therapeutic characteristics. It can significantly improve the quality of life for patients suffering from these diseases.

Since resveratrol contains polyphenols, it may have some potential health benefits for peri- and postmenopausal women. Its modulatory effects on ERα and ERβ and the impact of its polyphenol content as an epigenetic modifier on histones and methylation may lead to a profound improvement in menopause-related symptoms and/or disorders. A randomized, controlled, double-blind study reported that resveratrol was effective in reducing the number and intensity of hot flashes in 78.6% of patients [61]. In an RCT, resveratrol supplementation was reported to improve osteoarthritis-related pain perception and menopause-related quality of life, including mood, depressive symptoms, and sleep quality, in postmenopausal women [10].

Additionally, resveratrol enhances bone mineral density by stimulating osteoblastic activity while inhibiting osteoclastic activity in postmenopausal women [62]. In a 24-month randomized, double-blind, placebo-controlled crossover study, a 75 mg dose of resveratrol administered twice daily was found to enhance cognition, cerebrovascular function, and insulin sensitivity in postmenopausal women [25]. A recent review also suggested that chronic resveratrol intake may have a positive effect on brain function [63].

Most studies examining the effects of resveratrol on menopause are conducted in animals. RCTs that include postmenopausal women who have been given resveratrol for a long time are needed. Regarding the effectiveness and safety of resveratrol use in postmenopausal women, more evidence is required.

This review presents several limitations, most notably the scarcity of studies investigating the clinical use of resveratrol in estrogen-dependent conditions such as endometriosis, adenomyosis, lipedema, PCOS, and estrogen-receptor-positive breast cancer. This gap may be explained by the biological complexity of these disorders, which involve multifactorial hormonal and inflammatory pathways. Resveratrol exhibits dual activity on estrogen receptors (ERα and ERβ), acting as an agonist or antagonist depending on tissue specificity, concentration, and receptor subtype, which adds further complexity to clinical research. Additionally, the compound has poor oral bioavailability and is rapidly metabolized, while the biologically active trans isomer is chemically unstable, and few studies have employed standardized or bioavailable formulations. Most trials conducted to date also show significant methodological heterogeneity, including small sample sizes, non-standardized dosing regimens, short durations, and variability in patient characteristics. As a naturally occurring, non-patentable compound, resveratrol attracts limited commercial interest, which reduces funding opportunities for large-scale clinical trials. Moreover, although resveratrol is generally well-tolerated at doses of up to 1 g/day, safety concerns regarding potential estrogenic stimulation in hormone-sensitive tissues remain a barrier to long-term studies in populations with hormone-sensitive conditions. Therefore, further investigation through well-powered, rigorously designed randomized controlled trials is essential to define the clinical role of resveratrol in the management of inflammatory and estrogen-dependent conditions.

## 4. Conclusions

The review highlights the promising antioxidant, anti-inflammatory, and hormone-modulating properties of resveratrol, supporting its potential use as an adjuvant in the management of metabolic and estrogen-dependent conditions. These effects may contribute to improved symptom control and quality of life, particularly in pre- and postmenopausal women. Nevertheless, the current body of evidence remains limited, with a predominance of preclinical studies and a small number of clinical trials in humans. Existing studies are often heterogeneous in design, with limited sample sizes, short intervention periods, and a lack of standardized dosing protocols or clinically relevant endpoints.

Given these limitations, future randomized controlled trials are warranted to establish the efficacy and safety of resveratrol in these populations. Such studies should include well-characterized cohorts of women diagnosed with estrogen-dependent or metabolic conditions, stratified according to menopausal status. Intervention durations should be sufficiently long—ideally twelve weeks or more—to capture sustained metabolic and hormonal effects. Moreover, trials should employ standardized formulations of resveratrol, with clear specification of the isomer used (particularly trans-resveratrol) and any bioavailability-enhancing strategies, such as nanoencapsulation. Outcome measures should include not only biochemical and hormonal markers (such as E2, testosterone, SHBG, HOMA-IR, and lipid profile) but also validated clinical endpoints, including symptom severity scores, quality of life assessments, and inflammatory biomarkers. Importantly, safety profiles should be rigorously monitored, especially in populations with hormone-sensitive conditions, to assess potential proliferative effects on breast, endometrial, or ovarian tissues.

In conclusion, while resveratrol shows promise as a therapeutic agent, the current evidence remains insufficient to support its routine clinical use. Well-designed, large-scale clinical trials are crucial for determining the optimal dosing, long-term effects, and therapeutic relevance of resveratrol supplementation in metabolic and estrogen-dependent disorders.

## Figures and Tables

**Figure 1 cimb-47-00692-f001:**
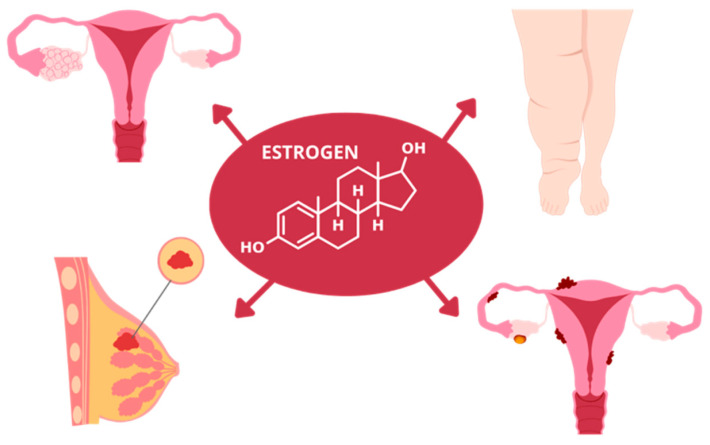
Relationship between estrogen and clinical conditions associated with estrogen disruption and ER activation. The figure illustrates the estrogen molecule at the center, representing its central role in the pathophysiology of various estrogen-dependent conditions. Arrows indicate the connection between estrogen and four disorders: PCOS (**top left**), adenomyosis/endometriosis (**bottom right**), lipedema (**top right**), and breast cancer (**bottom left**). The image highlights the systemic impact of estrogen signaling, reinforcing its relevance as a therapeutic target in multiple gynecological and metabolic contexts. ER: estrogen receptor; PCOS: polycystic ovarian syndrome.

**Figure 2 cimb-47-00692-f002:**
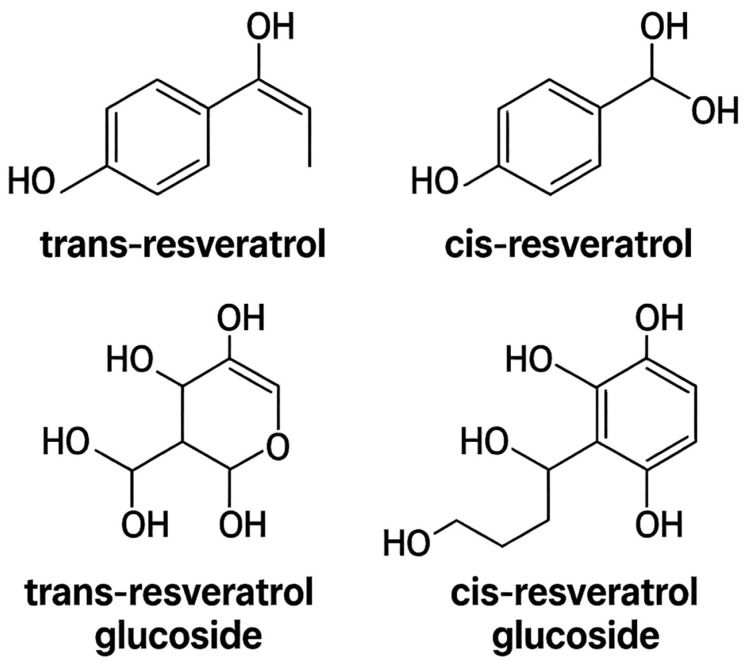
Chemical structures of the main isomeric and glycosylated forms of resveratrol. The upper panel displays trans-resveratrol (the more stable and biologically active form) and cis-resveratrol (the less stable and less biologically active form). The lower panel presents their respective glycosides: trans-resveratrol glucoside, which is predominant in plants, and cis-resveratrol glucoside. Glycosylation increases aqueous solubility but reduces bioavailability compared to that of the aglycone forms.

**Figure 3 cimb-47-00692-f003:**
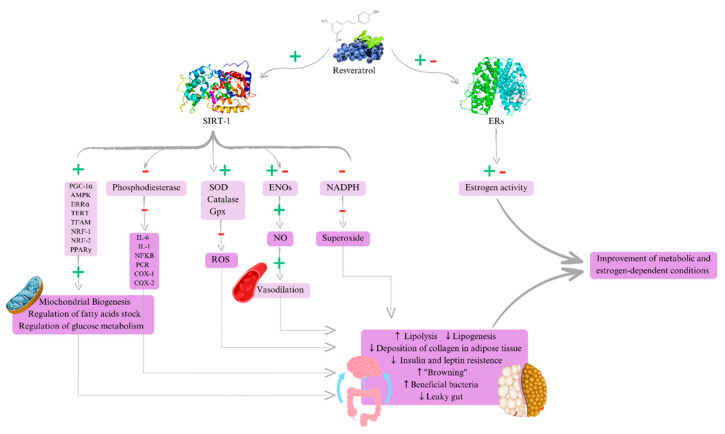
Overview of the role of resveratrol in the sirtuin and estrogen receptor pathways and their potential benefits for the treatment of metabolic and estrogen-dependent conditions. This schematic illustrates the molecular mechanisms through which resveratrol exerts beneficial effects via two primary pathways: activation of SIRT-1 (sirtuin 1) and modulation of estrogen receptors (ERs). On the left side, resveratrol positively modulates SIRT-1, a NAD^+^-dependent deacetylase involved in cellular stress responses and metabolic regulation. Activation of SIRT-1 leads to upregulation of key transcription factors and coactivators involved in mitochondrial biogenesis and metabolic homeostasis (PGC-1α, AMPK, ERRα, TERT, TFAM, NRF-1, NRF-2, and PPARγ), resulting in enhanced mitochondrial function, regulation of fatty acid storage, and improved glucose metabolism. Inhibition of phosphodiesterases reduces pro-inflammatory mediators (IL-6, IL-1, NFKB, PCR, COX-1, and COX-2), thus suppressing chronic inflammation. Increased expression of antioxidant enzymes (SOD, catalase, Gpx) reduces ROS (reactive oxygen species), thereby mitigating oxidative stress. Activation of endothelial nitric oxide synthase (ENO) leads to increased production of nitric oxide (NO) and consequent vasodilation, improving vascular health. Inhibition of NADPH oxidase reduces the production of superoxide, another major ROS source. On the right side, resveratrol also modulates estrogen receptors (ERs), exhibiting both agonistic and antagonistic effects depending on the tissue context. This modulation affects overall estrogenic activity, which is crucial in the pathogenesis of estrogen-dependent conditions. Downstream effects of these pathways include increased lipolysis and decreased lipogenesis; reduced collagen deposition in adipose tissue, contributing to reduced tissue fibrosis; improved insulin and leptin sensitivity; induction of adipose tissue “browning” (conversion to metabolically active brown-like adipocytes); enhanced intestinal barrier integrity, with reduction in leaky gut; and enrichment of beneficial gut bacteria. Collectively, these molecular and physiological effects contribute to the improvement of metabolic and estrogen-dependent conditions such as obesity, metabolic syndrome, and estrogen-driven disorders like lipedema, endometriosis, and others. Abbreviations: SIRT-1: sirtuin1; ERs: estrogen receptors; PGC-1α: peroxisome proliferator-activated receptor gamma coactivator-1 alpha; AMPK: AMP-activated protein kinase; ERRα: estrogen-related receptor alpha; TERT: telomerase reverse transcriptase; TFAM: mitochondrial transcription factor A; NRF-1: nucleoid related factor-1; NRF-2: nucleoid related factor-2; PPARγ: peroxisome proliferator-activated receptor gamma; IL-6: interleukin-6; IL-1: interleukin-1; NFKB: nuclear factor kappa B; PCR: C-reactive protein; COX-1: cyclooxygenase 1; COX-2: cyclooxygenase 2; SOD: superoxide dismutase; Gpx: glutathione peroxidase; ROS: reactive oxygen species; ENOs: endothelial nitric oxide synthase; NO: nitric oxide; NADPH: nicotinamide adenine dinucleotide phosphate. (+): positive regulation/activation; (−): negative regulation/inhibition; (+/–): dual modulation.

**Table 1 cimb-47-00692-t001:** Main dietary sources of resveratrol.

Source	Total Resveratrol (µg/100 g or µg/L)	Predominant Form(s)	[Reference]
Red grape	50–1000	*Trans* + Piceides	[7,10]
Red wine	1980–7130 µg/L	*Trans*	[7]
Raw peanut	74	*Trans* + *Cis*	[7]
Germinated peanut	1170–2570	*Trans*	[7]
Blueberry	50–100	*Trans*	[7]
Peach	461.6	Aglycone	[7]
Apple	67	*Cis + Trans*	[7]
Pear	34.43	*Cis + Trans*	[7]
Grapefruit	82	*Cis + Trans*	[7]

**Table 2 cimb-47-00692-t002:** Metabolic and anti-inflammatory actions of resveratrol in preclinical and clinical studies. GSH: reduced glutathione; Gpx: glutathione peroxidase; Hb1Ac: glycated hemoglobin; HDL: high-density lipoprotein cholesterol; HOMA-IR: Homeostatic Model Assessment for Insulin Resistance; IL-6: interleukin-6; IL-1: interleukin-1; MCP-1: monocyte chemoattractant protein-1; MDA: malondialdehyde; NF-kB: nuclear factor kappa B; PCOS: polycystic ovarian syndrome; PCR: C-reactive protein; PGC-1α: peroxisome proliferator-activated receptor gamma coactivator-1 alpha; RCT: randomized controlled trial; RSV: resveratrol; SIRT-1: sirtuin1; SOD: superoxide dismutase; T2DM: type 2 diabetes mellitus; TC: total cholesterol; TG: triglycerides; TNFα: tumoral necrose factor alpha; T-RSV: *trans*-resveratrol; VLDL: very-low-density lipoprotein cholesterol; UCP-1: uncoupling protein.

Reference	Sample	Duration	Study Design	Resveratrol Doses	Main Results
Abdollahi et al., 2019 [40]	71 overweight individuals with T2DM	8 weeks	Randomized double blind	T-RSV 1 g daily or placebo	Significative reduction in fasting blood glucose in the group that ingested T-RSV
Ashkar et al., 2020 [41]	10 Sprague–Dawley rats	21 days	Preclinical	Oral administration of 20 mg/kg/day resveratrol after PCOS induction	Improvement of lipid profile, insulin resistance, MDA, and TNF-α Increase in SOD Reduction in cystic follicles and ovarian weight
Batista-Jorge et al., 2020 [42]	25 obese patients with metabolic syndrome	12 weeks	Randomized trial	250 mg RSV or placebo capsule daily in combination with a physical activity program + diet for 3 months	RSV improved TC, HDL, VLDL, urea, creatinine, and albumin vs. placebo. Anthropometric parameters were significantly different after 3 months of physical activity for both placebo and RSV
Brenjian et al., 2020 [43]	Cumulus cells obtained from 40 patients with PCOS	40 days	Clinical	T-RSV 800 mg/kg/day orally	Alterations in serum Decrease in IL-1, IL-6, TNF-α, NF-κB
De Ligt et al., 2020 [44]	41 overweight individuals	24 weeks	Randomized double blind	T-RSV 150 mg daily or placebo	There was no improvement in insulin sensitivity, but there was a significant reduction in Hb1Ac in the group that ingested RSV
Du et al., 2020 [45]	18 male mice	28 days	Preclinical	Chitosan-encapsulate RSV 100 mg/kg/day orally	Decrease in blood glucose Decrease in TC, TG, LDL Increase in HDL Decrease in IL-6, MCP-1 Increase in SOD and GSH activity
Gorabi et al., 2021 [31]	1741 individuals	10 weeks	Meta-analysis of 35 RCTs	Average RSV and T-RSV ≥500 mg daily	Significant reduction in CRP in inflammatory conditions
Hoseini et al., 2019 [46]	56 individuals with T2DM and cardiovascular disease	4 weeks	Randomized double blind	500 mg daily or placebo	Significant effect on glycemic control in the intervention group
Andrade et al., 2019 [47]	32 male mice	8 weeks	Randomized clinical trial	T-RSV 500 mg daily	Increase gene expression of SIRT1 Increase in UCP-1 expression Increase in adiponectin levels Decrease in TC and TG levels
Mahjabeen et al., 2019 [48]	110 subjects with T2DM	24 weeks	Randomized, double-blind, placebo-controlled parallel-group trial	T-RSV 200 mg/day	Significant decrease in glucose and Hb1Ac Significant decrease in insulin and HOMA-IR
Sattarinezhad et al., 2019 [49]	60 individuals with T2DM and albuminuria	12 weeks	Randomized double blind	T-RSV 500 mg daily + losartan or placebo +losartan	Significant reduction in diabetes parameters like fasting blood glucose and insulin resistance
Thaung et al., 2021 [25]	125 postmenopausal women	24 months	Randomized double blind	75 mg daily or placebo	Improvement in insulin resistance in the group that ingested RSV
Zhao et al., 2019 [50]	30 male mice, 6 weeks old	12 weeks	Preclinical	60 mg/kg/day i.g.	Decrease in blood glucose Increase in insulin sensitivity Decrease in serum TC and hepatic TG
Wang et al., 2020 [51]	21 Sprague–Dawley mice, 3 weeks old	35 days	Preclinical	Daily injection of 100 mg/kg resveratrol during the PCOS induction	Reduction in body weight, ovarian interstitial fibrosis, and serum and ovarian levels of MDA Increase in ovarian weight, numbers of luteal cells and antral follicles, and serum and ovarian levels of SOD Enhancement of SIRT1 protein expression No effect on androgen receptor

## Data Availability

The datasets used and/or analyzed during the current study are available from the corresponding author upon reasonable request.

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
