# Peer review of "New Perspectives on the Use of Resveratrol in the Treatment of Metabolic and Estrogen-Dependent Conditions Through Hormonal Modulation and Anti-Inflammatory Effects"

_cimb, 2025, doi:10.3390/cimb47090692_

Round 1

Reviewer 1 Report

Comments and Suggestions for Authors

This study provides a comprehensive review of resveratrol’s potential therapeutic effects on metabolic and estrogen-dependent conditions, focusing on its hormonal modulation, anti-inflammatory properties, and bioavailability challenges.

Below, I outline a series of observations and questions for the authors to address in an organized manner:

  • Did any of the reviewed studies show dose–response relationships, particularly in estrogen dependent conditions and was resveratrol effective?
  • What are the differences in therapeutic efficacy of trans-resveratrol compared to other isomeric or formulated forms?
  • Have any side effects been reported in long-term supplementation studies?

Author Response

This study provides a comprehensive review of resveratrol’s potential therapeutic effects on metabolic and estrogen-dependent conditions, focusing on its hormonal modulation, anti-inflammatory properties, and bioavailability challenges.

Below, I outline a series of observations and questions for the authors to address in an organized manner:

  • Did any of the reviewed studies show dose–response relationships, particularly in estrogen dependent conditions and was resveratrol effective? Thank you for your attention to this point. Yes, we agree with the reviewer and we have incorporated the current evidence on dose–response and time–response relationships in treatments with resveratrol, along with possible side effects associated with its use, detailed in the eleventh, twelfth, and thirteenth paragraphs.
  • What are the differences in therapeutic efficacy of trans-resveratrol compared to other isomeric or formulated forms? Thank you for your attention to this point. Yes, we agree with the reviewer and have included additional information on the different existing forms of resveratrol and their observed therapeutic potential, as well as formulations with the capacity to enhance its bioavailability—specifically in the fourth and fifth paragraphs —in the Introduction section.
  • Have any side effects been reported in long-term supplementation studies? Thank you for your attention to this point. Yes, we agree with the reviewer and included side effects reported in clinical trials from lines 154 to 162. 

    Thanks you!

Reviewer 2 Report

Comments and Suggestions for Authors

New perspectives on the use of resveratrol in the treatment of metabolic and estrogen-dependent conditions through hormonal modulation and anti-inflammatory effects. cimb-3804034

This review propose to resveratrol as a potential agent for the treatment of metabolic and estrogen diseases. Also, show the state of the art in the period 2000 to 2024.

Abstract: is ok

Kerwords,  authors may include teh following suggestion:

Considering that review show that trans-resveratrol is the most important than cis-resveratrol change resveratrol as trans-resveratrol; delete estrogen, sirtuin and metabolism. Include anti-inflamamatory factors

Keywords: trans-resveratrol, estrogen receptor, metabolic sybdrome, anti-inflamatory factors

Introduction

Authors must standardize subtitles in the use of  capital letters or uppercase letters see for example lines 55 and 244. Please check all subtitules

Line 62 in vitro and in vivo are italics

Check all manuscript cis  or trans are in italics  ; lines 73, 75,79, 82, 93, 99

Lines 75, 86, 97, 105 change Resveratrol as resveratrol

Line 89 change capital letters for uppercase letters

Result and Discussion

Lines 130 to 144; 148 to 154 change capital letters for uppercase letters

Line 162 Enterococcus faecalis is in italics

Line 174 change Resveratrol as resveratrol

Line 253 trans is in italics

References

Check all refrences the name of the journals is abbreviation; also, change capital letters for uppercase letters

Author Response

New perspectives on the use of resveratrol in the treatment of metabolic and estrogen-dependent conditions through hormonal modulation and anti-inflammatory effects. cimb-3804034

This review propose to resveratrol as a potential agent for the treatment of metabolic and estrogen diseases. Also, show the state of the art in the period 2000 to 2024.

Abstract: is ok. Thank you!

Kerwords,  authors may include teh following suggestion: Considering that review show that trans-resveratrol is the most important than cis-resveratrol change resveratrol as trans-resveratrol; delete estrogen, sirtuin and metabolism. Include anti-inflamamatory factors. Keywords: trans-resveratrol, estrogen receptor, metabolic sybdrome, anti-inflamatory factors. Thank you! We adjusted all keywords according to the reviewer's suggestions.  

Introduction Authors must standardize subtitles in the use of  capital letters or uppercase letters see for example lines 55 and 244. Please check all subtitules. Line 62 in vitro and in vivo are italics. Check all manuscript cis  or trans are in italics  ; lines 73, 75,79, 82, 93, 99. Lines 75, 86, 97, 105 change Resveratrol as resveratrol. Line 89 change capital letters for uppercase letters. Thank you for your comments. I have incorporated all the requested modifications throughout the text . I would also like to mention that, as additional revisions to the text were suggested, some significant structural changes were made to the article's Introduction section, fulfilling all reviewer requests. 

Result and Discussion. Lines 130 to 144; 148 to 154 change capital letters for uppercase letters. Line 162 Enterococcus faecalis is in italics. Line 174 change Resveratrol as resveratrol. Line 253 trans is in italics. References: Check all refrences the name of the journals is abbreviation; also, change capital letters for uppercase letters. Thank you for your comments. I have incorporated all the requested modifications throughout the text and references. I would also like to mention that, as additional revisions to the text were suggested, some significant structural changes were made to the article's Introduction section, fulfilling all reviewer requests. 

Reviewer 3 Report

Comments and Suggestions for Authors

The manuscript provides a comprehensive review of resveratrol's potential benefits in metabolic and estrogen-dependent conditions, supported by a thorough analysis of existing literature. The topic is timely and relevant, given the increasing interest in natural compounds for managing hormonal and metabolic disorders.

Major Comments:

    • While the manuscript discusses resveratrol's effects on SIRT-1 and estrogen receptors, a more detailed mechanistic explanation of how resveratrol modulates these pathways would strengthen the review. For example, how exactly does resveratrol's interaction with ERα and ERβ lead to anti-inflammatory or metabolic benefits?
    • The manuscript mentions issues with resveratrol's bioavailability but does not provide a clear consensus on optimal dosages or formulations for clinical use. A dedicated section summarizing current knowledge on effective doses and delivery methods would be beneficial.
    • The review highlights the lack of randomized controlled trials (RCTs) in estrogen-dependent conditions. It would be helpful to discuss potential reasons for this gap and suggest specific study designs or populations for future research.
    • The figure is informative but could be improved with clearer labels and a more detailed legend to explain the pathways and interactions depicted.

Minor Comments:

    • Some sentences are overly complex or could be streamlined for better readability. For example, the phrase "Its signs and symptoms are often confused, requiring careful attention by the specialist to diagnose" could be reworded for clarity.
    • Ensure all citations are consistently formatted and complete. For instance, some references in the text lack corresponding entries in the reference list (e.g., citation numbers 1-3 in the Introduction).
    • There are minor typographical errors (e.g., "Estrogen-dependent conditions, such as endometriosis, adenomyosis, lipedema, and breast cancer, are intimately involved with hormonal changes related to estrogen and its receptors." – "its" should be "their").
    • The conclusion briefly mentions the need for larger RCTs. Expanding this section to include specific recommendations for future research (e.g., study duration, patient populations, or outcome measures) would add value.
    • Consider discussing potential synergies between resveratrol and existing therapies and the potential nanoformulation for enhancing the therapeutic effect (https://doi.org/10.3390/pharmaceutics17010114; https://doi.org/10.3390/biochem4010003).

Author Response

Major Comments:

    • While the manuscript discusses resveratrol's effects on SIRT-1 and estrogen receptors, a more detailed mechanistic explanation of how resveratrol modulates these pathways would strengthen the review. For example, how exactly does resveratrol's interaction with ERα and ERβ lead to anti-inflammatory or metabolic benefits? Thank you for your attention to this point. We added more in-depth information about resveratrol’s effects on estrogen receptors in the introduction, according to your suggestions. 
    •  
    • The manuscript mentions issues with resveratrol's bioavailability but does not provide a clear consensus on optimal dosages or formulations for clinical use. A dedicated section summarizing current knowledge on effective doses and delivery methods would be beneficial. Thank you for your attention to this point. We have added more in-depth information about resveratrol’s doses and delivery methods in the introduction, as reviewer suggestions. 
    •  
    • The review highlights the lack of randomized controlled trials (RCTs) in estrogen-dependent conditions. It would be helpful to discuss potential reasons for this gap and suggest specific study designs or populations for future research. Thank you for your attention to this point. We have added more in-depth information about the reasons for the lack of clinical trials with humans involving resveratrol in the introduction, as reviewer suggestions. 
    •  
    • The figure is informative but could be improved with clearer labels and a more detailed legend to explain the pathways and interactions depicted. Thank you for your attention to this point. We have revised the figure and also added another one according to the reviewer's suggestions. 

Minor Comments:

    • Some sentences are overly complex or could be streamlined for better readability. For example, the phrase "Its signs and symptoms are often confused, requiring careful attention by the specialist to diagnose" could be reworded for clarity. Thank you for your attention to this point. We adjusted according to the reviewer's suggestions. 
    •  
    • Ensure all citations are consistently formatted and complete. For instance, some references in the text lack corresponding entries in the reference list (e.g., citation numbers 1-3 in the Introduction). Thank you for your attention to this point. We adjusted all citations according to the reviewer's suggestions. 
    •  
    • There are minor typographical errors (e.g., "Estrogen-dependent conditions, such as endometriosis, adenomyosis, lipedema, and breast cancer, are intimately involved with hormonal changes related to estrogen and its receptors." – "its" should be "their"). Thank you for your attention to this point. We adjusted in the abstract according to the reviewer's suggestions. 
    •  
    • The conclusion briefly mentions the need for larger RCTs. Expanding this section to include specific recommendations for future research (e.g., study duration, patient populations, or outcome measures) would add value. Thank you for your attention to this point. We have adjusted the conclusion in accordance with the reviewer's suggestions, adding specific recommendations for future research. 
    •  
    • Consider discussing potential synergies between resveratrol and existing therapies and the potential nanoformulation for enhancing the therapeutic effect (https://doi.org/10.3390/pharmaceutics17010114; https://doi.org/10.3390/biochem4010003). Thank you for your attention to this point. We don't agree with the reviewer on this matter, and we didn't include these references because they don't fit the purpose of our review. 

Thank you for the feedback regarding the article. I have taken all considerations into account to improve the manuscript. Best regards

Reviewer 4 Report

Comments and Suggestions for Authors

Estrogen-dependent conditions are associated with alterations in estrogen function and inflammatory mechanisms, however resveratrol can treat inflammatory diseases like obesity, metabolic syndrome, and endometriosis. This review reveals that resveratrol may benefit metabolic and estrogen-dependent conditions by modulating anti-inflammatory factors that regulate estrogen receptor activity, increasing lipolysis, decreasing insulin resistance, and mitigating oxidative stressed. It has great theoretical significance and high practical value of  therapeutic effects of resveratrol in metabolic conditions. Although the article is of interest and comprehensive, and meets the Research Topic in general, smooth language and so on. However, there are some major issues need to be improved:

  1. Abstract: The abstract should be modified to highlight the innovation points and avoid duplication;
  2. Introduction: There are few references, and the impact of global metabolic diseases and esthenosis on human health, dfferences in resveratrol content in plants, specially the latest research progress of resveratrol in the treatment of metabolic diseases and esthenosis should be supplemented; The title of section 1.1 could be omitted, but the level and logic of the paragraphs needed improvement;
  3. Materials and Methods: Too many screening keywords and few literature
  4. Results and Discussion: The table is too small and the layout of the table is not reasonable. It is suggested to modify it. You can refer to https://www.mdpi.com/1420-3049/29/13/3110; There are too few pictures. Can we add more?Results and discussion can only be divided into two parts; The results focus on the conclusions of the 48 references identified and summarized; There should be a large number of references in the discussion, except 48 references in results.
  5. References: There are few references. If we follow the existing literature, it can only be considered as a mini review.

Author Response

Estrogen-dependent conditions are associated with alterations in estrogen function and inflammatory mechanisms, however resveratrol can treat inflammatory diseases like obesity, metabolic syndrome, and endometriosis. This review reveals that resveratrol may benefit metabolic and estrogen-dependent conditions by modulating anti-inflammatory factors that regulate estrogen receptor activity, increasing lipolysis, decreasing insulin resistance, and mitigating oxidative stressed. It has great theoretical significance and high practical value of  therapeutic effects of resveratrol in metabolic conditions. Although the article is of interest and comprehensive, and meets the Research Topic in general, smooth language and so on. However, there are some major issues need to be improved:

  1. Abstract: The abstract should be modified to highlight the innovation points and avoid duplication; 

    We thank the reviewer for the attention to this point. The abstract was revised to eliminate unnecessary repetition and to emphasize the most important points. We also adjusted the keywords.

  2. Introduction: There are few references, and the impact of global metabolic diseases and esthenosis on human health, dfferences in resveratrol content in plants, specially the latest research progress of resveratrol in the treatment of metabolic diseases and esthenosis should be supplemented; The title of section 1.1 could be omitted, but the level and logic of the paragraphs needed improvement;

We thank the reviewer for the attention to this point. In the first three paragraphs of the Introduction, we added content highlighting the importance of identifying new ways to treat metabolic and estrogen-dependent diseases, given their potential impact on global health. In paragraphs 4 to 6, we included more relevant information about dietary sources of resveratrol, as well as its different isomers and biological potentials. Furthermore, I improved the overall organization of the text and removed the subsection title, as recommended.

  1. Materials and MethodsToo many screening keywords and few literature

We thank the reviewer for the attention to this point. In the Materials and Methods section, the keywords were reduced to only those most relevant to the study.

  1. Results and DiscussionThe table is too small and the layout of the table is not reasonable. It is suggested to modify it. You can refer to https://www.mdpi.com/1420-3049/29/13/3110; There are too few pictures. Can we add more?Results and discussion can only be divided into two parts; The results focus on the conclusions of the 48 references identified and summarized; There should be a large number of references in the discussion, except 48 references in results.

We thank the reviewer for the attention to this point. The table was reformatted according to the suggested layout, keeping only the most relevant information to enhance clarity and appropriateness to the article. Three additional figures were created to illustrate: (1) the main dietary sources of resveratrol, (2) the principal isoforms of resveratrol, and (3) the main estrogen-dependent health conditions. Additionally, I created a table showing the average amounts of resveratrol in various plant-based foods in order to compare them with the doses typically used in supplementation or targeted treatments.

  1. References: There are few references. If we follow the existing literature, it can only be considered as a mini review.

We thank the reviewer for the attention to this point. To address the requested additions and further enrich the depth of the topic, I also searched for new references, which are listed until reference 63.

Thank you!

Round 2

Reviewer 3 Report

Comments and Suggestions for Authors

The authors have satisfactorily addressed all of my previous comments. I would, however, recommend adding the corresponding NCT number of the clinical trial in Table 2, if available. After this minor adjustment, the manuscript can be accepted.

Author Response

The authors have satisfactorily addressed all of my previous comments. I would, however, recommend adding the corresponding NCT number of the clinical trial in Table 2, if available. After this minor adjustment, the manuscript can be accepted.

Response: We thank the reviewer for the attention to this point. We agree with the reviewer and have added more detailed reference sources to Table 2, following the authors' guidelines. 

Reviewer 4 Report

Comments and Suggestions for Authors

The author has revised the paper strictly according to the review comments and the quality of the paper has been greatly improved. I agree to accept the publication

Author Response

The author has revised the paper strictly according to the review comments and the quality of the paper has been greatly improved. I agree to accept the publication

Response: We thank the reviewer.

Best regards